# Progenitor/Stem Cells in Vascular Remodeling during Pulmonary Arterial Hypertension

**DOI:** 10.3390/cells10061338

**Published:** 2021-05-28

**Authors:** France Dierick, Julien Solinc, Juliette Bignard, Florent Soubrier, Sophie Nadaud

**Affiliations:** 1Lady Davis Institute for Medical Research, McGill University, Montréal, QC H3T 1E2, Canada; france.dierick@mail.mcgill.ca; 2UMR_S 1166, Faculté de Médecine Pitié-Salpêtrière, INSERM, Sorbonne Université, 75013 Paris, France; julien.solinc@inserm.fr (J.S.); juliette.bignard@gmail.com (J.B.); florent.soubrier@sorbonne-universite.fr (F.S.)

**Keywords:** pulmonary arterial hypertension, vascular remodeling, progenitor cells, stem cells, endothelial cells, smooth muscle cells, pericytes

## Abstract

Pulmonary arterial hypertension (PAH) is characterized by an important occlusive vascular remodeling with the production of new endothelial cells, smooth muscle cells, myofibroblasts, and fibroblasts. Identifying the cellular processes leading to vascular proliferation and dysfunction is a major goal in order to decipher the mechanisms leading to PAH development. In addition to in situ proliferation of vascular cells, studies from the past 20 years have unveiled the role of circulating and resident vascular in pulmonary vascular remodeling. This review aims at summarizing the current knowledge on the different progenitor and stem cells that have been shown to participate in pulmonary vascular lesions and on the pathways regulating their recruitment during PAH. Finally, this review also addresses the therapeutic potential of circulating endothelial progenitor cells and mesenchymal stem cells.

## 1. Introduction

The origin of the vascular remodeling occurring during the various forms of pulmonary arterial hypertension (PAH) is still unexplained, although some pathological conditions or gene defects are known to favor the development of the disease. This remodeling can predominate in the arterial compartment in PAH or the venous side in pulmonary veno-occlusive disease (PVOD) through the production of new endothelial cells (EC), myofibroblasts, vascular smooth muscle cells (SMC), and also through extracellular matrix changes with intimal and medial fibrosis in the intima [1]. It leads to vessel narrowing and stiffening and ultimately to complex occlusive vascular lesions, called plexiform lesions, containing proliferative and apoptosis-resistant EC. In order to elucidate the mechanisms of the remodeling, advances have been made to identify the cellular processes leading to endothelium and medial cell proliferation. Because it is difficult to explain the pathological process solely by in situ proliferation of resident vascular cells, the contribution of extra-vascular cells invading the vascular wall was explored. These vascular progenitor and stem cells are undifferentiated cells that can produce new vascular cells and are characterized by their self-renewal capacities and ability to form colonies (colony-forming unit or CFU). Their differentiation potential varies from large in stem cells, such as mesenchymal stem cells, to low in progenitor cells limited to a few lineages. Identification of the invading cells is important to give insight into the active pathological process and to design treatments for the disease by inhibiting this process. These cells can be either mobilized from cellular niches located within the vessel wall, or in its vicinity, in the lung interstitium, or from distant tissues, mainly the bone marrow, through the circulation to differentiate in their tissular target destination (Figure 1). Additional mechanisms of cell trans-differentiation or cell transition can be involved that are able to change the vessel morphology. However, identifying the cell population involved in the remodeling does not provide the primum movens of the remodeling as their recruitment is under the control of signaling pathways regulated by ligands bound to their cognate receptors. Indeed, these cellular mechanisms can be shared among different types of PAH of various origins. As such, the Bone Morphogenetic Proteins (BMP) paradigm is an important clue for deciphering the pathological scenario as this pathway is downregulated in subjects carrying a heterozygous loss of function mutation of the BMP receptor type 2 gene (*BMPR2*) but also in *BMPR2* mutation non-carrier PAH patients. This signaling pathway dysregulation plays a major role, and some studies have indeed demonstrated that it also controls progenitor and stem cell behavior. Thus recent results of a protector effect of a BMP treatment in *Bmpr2* mutated mice [2] or of a proteic compound acting as a Transforming Growth Factor β (TGF-β) decay molecule designed to counterbalance the deficient BMP pathway [3] could also involve regulation of vascular progenitor and stem cells.

Here, we will review the various types of progenitor and stem cells that have been shown to be involved in pulmonary vascular remodeling during pulmonary hypertension (PH) and the signaling pathways that can modulate their recruitment.

## 2. Endothelial Progenitor Cells

Almost a quarter of a century ago, in *Science*, Asahara and colleagues first reported the identification and purification of circulating endothelial progenitor cells (EPC) based on two cell surface antigens, CD34 and Vascular Endothelial Growth Factor Receptor 2 (VEGFR-2) [4]. These cells are derived from the postnatal bone marrow (BM), enter the peripheral blood, differentiate into endothelial cells (EC), and participate in the formation of new blood vessels or repair the damaged EC into mature resident vessels [5]. Since EPC and hematopoietic stem cells share a common embryonic precursor, these cells could share several markers, such as c-kit, CD133, Sca-1 (in mouse), VE-cadherin (Vascular Endothelial cadherin), VEGFR-2, and CD105, which can make their characterization complicated. So far, two distinct subsets of EPCs have been well-described: the early outgrowth EPC are derived from the hematopoietic lineage and originate from the BM, and the late outgrowth EPC are derived from endothelial lineage and could arise from tissue vascular niches [6,7,8]. Indeed, in postnatal life, early progenitor phenotype could be represented by CD133^+^/CD34^+^/VEGFR-2^+^ cells and are located mainly in the BM. Then, during their differentiation, EPC lose CD133 and start to express CD31, VE-cadherin, and von Willebrand Factor (vWF). These more mature EPC are found in the peripheral circulation and within specific regions of the adult blood vessel wall, the subendothelial zone of the intima, or in the vasculogenic zone between the media and adventitial layers (for review on EPC, see [9]). In addition, some of these resident vascular progenitor cells possess a bilineage potential and are also able to differentiate into vascular smooth muscle cells, both in vitro and in vivo [10].

Many endothelial progenitor cell populations have been identified in the adult lung, in particular in the context of pulmonary hypertension (PH). Studies have displayed contrasted results on the regulation of circulating EPC during PAH by measuring blood cells expressing different combinations of the markers CD133, CD34, and VEGFR-2. Circulating EPC were found reduced in some cases inversely correlated with the increase in the mean pulmonary arterial pressure (mPAP) [11,12,13,14,15,16]. However, others have shown that an elevated number in human idiopathic PAH and pulmonary fibrosis lung samples could be correlated with the severity of the disease and high pulmonary arterial pressure [17,18,19,20,21]. These results suggested a compensatory EPC proliferation in patients with end-stage PAH [22], and the higher the mPAP is, the more EPCs are consumed to repair pulmonary endothelium, causing a reduction in the number of EPCs [13]. CD133^+^ or CD133^+^/KDR^+^ cells were also found within arterial lesions in PAH patients suggesting that these EPCs may indeed participate in the vascular remodeling process by generating new EC [17,18,21]. EC in vascular lesions are characterized by a hyperproliferative and apoptosis-resistant phenotype in PH patients compared with control lung tissues [23,24]. Functional studies using late-outgrowth progenitor cells from familial PAH patients (with *BMPR2* gene mutations) demonstrated the presence of hyperproliferative endothelial phenotype (CD45^+^/CD133^+^/c-kit^+^/CXCR4^+^) in remodeled pulmonary arteries, occlusive and plexiform lesions, with impaired ability to form vascular networks and altered BMPR2 pathway [18]. These observations are similar to the phenotypic descriptions of alterations in pulmonary ECs that have been previously established in PAH [11,25]. Using the chronic hypoxia (CH)-induced PH model, studies showed an increase in the number of circulating EPCs in hypoxic animals [26] and observed that the neovascularization capacity of EPCs from hypoxic mice was impaired compared with EPCs from controls [27]. In addition, several studies suggested that circulating BM-derived progenitor cells (EPC and c-kit^+^) are deleterious and contribute to the disease pathogenesis. Mice transplanted with BM-derived CD133^+^ progenitor cells from patients with PAH, but not from healthy controls, exhibited morbidity and/or death due to features of PAH [28]. BM c-kit^+^ cells from CH animals were found hyperproliferative, and their transplantation into control animals promoted occlusive pulmonary arteriopathy in rats under exposure to CH [29]. They contribute to pulmonary vascular remodeling in the adventitia and the vasa vasorum of pulmonary arteries [30], and their recruitment was found to depend on serotonin 5-HT2B receptor expression [31].

Other works revealed that mouse tissue-resident EPC predominantly contribute to pulmonary vascular repair after endotoxin-induced injury [32] or in response to hypoxia [27,33] compared with BM-derived cells. Indeed, in adult rat lung, a vasculogenic resident microvascular endothelial cell population expressing endothelial cell markers (CD31, CD144, endothelial Nitric Oxide Synthase -eNOS-, or vWF), progenitor cell antigens (CD34 and CD309) and negative for CD45 has been isolated [34]. Nijmeh et al. described a high proliferative potential colony-forming endothelial progenitor-like cells in the calf pulmonary artery adventitial vasa vasorum. These cells are more numerous in the hypoxic calf and have higher expression levels of CD31, CD105, and c-kit than normoxic animals [35]. In human pulmonary vasculature, the presence of endothelial progenitor cell population was also described, expressing c-kit [36], CD31, vWF, eNOS, CD34, and caveolin-1 markers [37]; they are more proliferative and numerous in PAH patients.

## 3. SMC Progenitor Cells (SPC)

Resident SPC were identified through their involvement in vascular neointimal lesions during atherosclerosis or after endothelium injury. They also show an important role in ischemia-induced vascular regeneration (for review, see [38]). These resident cells have been characterized using a large variety of markers which makes it difficult to compare between the various studies. However, lineage-tracing experiments have given new insights into their differentiation potential.

Neomuscularization of normally non-muscularized pulmonary arterioles and abnormal muscularization of partially muscularized vessels are characteristic of PAH [1]. One of the major players in this process is the SMCs present in the media of pulmonary arteries. Adult vascular SMCs are quiescent, contractile, highly differentiated and specialized cells, identified by the combined expression of specific markers, such as α-Smooth Muscle Actin (α-SMA), Smooth Muscle-Myosin Heavy Chain (SM-MHC), Smooth Muscle 22α (SM22α), h1-calponin, and smoothelin. During PAH, SMCs can undergo a phenotypic transition process switching from a quiescent contractile state to a proliferative, migratory, and/or secretory state, associated with decreased expression of their specific marker genes [39,40,41]. This phenotypic switch is dependent on the microenvironment and partly related to endothelial dysfunction. An imbalance between vasodilatory, antiproliferative factors (nitric oxide and prostacyclin) and vasoconstrictive pro-proliferative molecules (endothelin-1, angiotensin II, and thromboxane) enhances SMC contractility but also stimulates their proliferation leading to the accumulation of dedifferentiated SMC in the neointima of remodeled vessels. In addition, other EC-derived growth factors (Platelet-Derived Growth Factor or PDGF, Vascular Endothelial Growth Factor or VEGF, Fibroblast Growth Factor or FGF, Interleukin 6 or IL-6) will, in turn, stimulate the proliferation and migration of SMC. The increased EC expression of Stromal cell-Derived Factor 1 (SDF-1) and Macrophage Migration Inhibitory Factor (MIF) can induce marked FoxM1 expression in SMC, inducing their proliferation as well [1,42]. Moreover, the reduced BMPR2 signaling also leads to increased SMC proliferation [43].

A small population of SMC primed to proliferate was identified in pulmonary arterioles, localized close to the muscularized-non-muscularized zone border [44]. Upon CH, these α-SMA^+^/SM-MHC^+^/ PDGF receptor type β (PDGFRβ)^+^ SMC follow a sequential program of dedifferentiation/redifferentiation to spread along the initially non-muscularized zone. This transition is characterized by a decrease in SM-MHC expression (terminal marker of differentiation) and the expression of the pluripotential factor Kruppel-Like Factor 4 (KLF4) induced by the activation of PDGFRβ following the increase in pulmonary EC-derived PDGF-B production. Increased KLF4 expressing PDGFRβ^+^/KLF4^+^ SMC were found in remodeling pulmonary arteries in the CH mouse model as well as in pulmonary arterioles of PH and PAH patients with a strong correlation with proliferative SMCs [44,45,46].

The new SMCs observed during neomuscularization can also originate from other stem/progenitor cells. Pericytes are mural cells that strongly interact with endothelial cells maintaining the structure of pre-capillary arterioles, capillaries, and post-capillary venules. They have been proposed as a source of SMC-like cells (for review on pericytes and fibroblasts, see [47]) and are considered to be close to multipotent mesenchymal stem cell (MSC) as they both are perivascular and display large osteogenic, chondrogenic, and adipogenic differentiation capacities (for review on MSC, see [48]). The heterogeneity of these multipotent cells does not allow the definition of specific markers, but their expression profile often involves markers such as PDGFRβ, NG2 Neural/Glial Antigen 2), CD146, Regulator Of G Protein Signaling 5 (RGS5), 3G5. They have been defined in alveolar regions as NG2^+^/PDGFRβ^+^/α-SMA^−^ [47] but also as α-SMA^−^/CD34^−^/CD31^−^/CD146^+^ in the lung and other organs [49,50]. Involvement of pericytes in PAH was first suggested by Meyrick and Reid in 1980, then by Patel et al., who proposed that a pericyte recruitment deregulation was the cause of the development of PH in two infants with Adams–Oliver syndrome [51,52]. Increased NG2^+^/3G5^+^ pericyte coverage of pulmonary vessels was indeed demonstrated in PH models induced by CH or monocrotaline (MCT) injection with early recruitment of pulmonary pericytes participating in vascular remodeling, as demonstrated using elegant lineage tracing models [53,54,55]. The increase in pericyte coverage is also found in the lungs of patients with idiopathic PAH (iPAH) and heritable PAH (hPAH) patients with a low proportion of α-SMA^+^/SM22^+^ pericytes in contrast to pericytes from control lungs, which are negative for these two SMC markers [54]. Moreover, SMCs and pericytes from PAH patients showed significantly overlapping transcriptomic profiles compared to healthy donors [55].

Other resident pulmonary MSC with a potential to differentiate into SMC-like cells have also been identified. The side population of progenitor cells, characterized by ATP Binding Cassette Subfamily G Member 2 (ABCG2) expression, are clonogenic and multipotent and co-express other stem/progenitor cell markers, such as c-kit, Sca-1, CD34 [56]. In vitro, they are able to differentiate into α-SMA^+^ SMC/myofibroblasts and NG2^+^ pericytes. Moreover, in vivo, when activated by increased reactive oxygen species, they contribute to CH-induced pulmonary vascular remodeling by adopting a contractile phenotype (α-SMA^+^) [56]. More recently, our team has also identified several cell populations of Paternally Widely 1 (PW1)-expressing resident mesenchymal progenitor cells present in both mice and humans. These progenitor cells proliferate and differentiate into SMC as early as after 4 days of CH, contributing to the muscularization and neomuscularization of pulmonary vessels [50]. The early recruited progenitor cells co-express PW1, CD34, Sca-1, PDGF Receptor type α (PDGFRα), and for some of them, c-kit. Their role in human pulmonary vascular remodeling is suggested by increased numbers of PW1^+^ perivascular cells and by the presence of PW1^+^/α-SMA^+^ medial SMC in iPAH patient lungs.

EC are able to undergo an endothelial–mesenchymal transition (EndMT) and to transdifferentiate into SMC-like [57]. Both in vitro and in vivo experiments, including elegant lineage tracing studies, have shown that ECs can adopt during PH a transient phenotype with co-expression of endothelial and SMC markers [58,59,60,61]. Since decreased BMPR2 expression was consistently observed in PAH, Ranchoux et al. established a genetically modified rat model with a heterozygous *Bmpr2* mutation and showed that EndMT and BMPR2 signaling alteration are linked and involved in advanced PH lesions [62]. EC undergoing EndMT start to express typical factors such as Twist [62], Slug, Snail [60], followed by expression of mesenchymal genes (α-SMA, vimentin) and even SMC specific proteins SM-MHC [58]. In human PAH lung samples, the presence of α-SMA^+^ cells expressing endothelial markers, such as VE-cadherin, CD31, or vWF, in the intima and neointima strongly supports that a neointimal cell population could originate from the endothelial cell lineage [58,59,60,62]. EndMT is promoted by inflammatory cytokines, such as TGF-β, Tumor Necrosis Factor-α (TNF-α), Interleukin 1β (IL-1β), and also by endothelin-1 [59]. In addition, Hopper et al. have observed that BMPR2 expression reduction in pulmonary artery endothelial cells (using siRNA in vitro and EC-specific *Bmpr2* knockout mice in vivo) triggers an increase in High Mobility Group AT-hook 1 (HMGA1) factor and smooth muscle cell protein expressions (α-SMA, SM22α, h1-calponin), and a decrease in CD31 expression, meaning that EndMT is induced by the loss of BMPR2 requiring the HMGA1 factor in PAH [63].

Bone marrow transplantation experiments also demonstrated the recruitment of circulating stem/progenitor cells participating in pulmonary vascular remodeling during PH. Indeed, lungs from CH chimeric mice show an accumulation of BM-derived cells, expressing α-SMA for some of them, indicating the acquisition of a SMC-like profile [64,65]. However, the role of recruited progenitor cells is somewhat disputed as it was less consistent in some studies [27,50,66,67]. Currently, there are no consensus markers for these circulating SMC progenitor cells, but c-kit is commonly used to define them. In hypoxic lungs, an accumulation of c-kit^+^/Sca-1^+^/α-SMA^+^ cells [30,68,69] as well as VEGFR-2^+^/c-kit^+^ cells [26] has been observed in vessels. C-kit^+^ cells were among the first stem/progenitor cells to be identified in and around vessels, and c-kit is considered as a stem cell marker. Resident c-kit^+^ progenitor cells are found in the adventitia of large vessels where they can produce new SMC participating in regenerating the vascular wall after injury [70]. One major difficulty with the c-kit marker is that it is also expressed on bone marrow-derived cells that are found in tissues as CD45^+^/c-kit^+^ cells. These circulating cells have been shown to be recruited in the adventitia of lung vessels in PAH models as Sca-1^+^/c-kit^+^ cells [68]. In vitro, they can differentiate into EC and SMC [30,71]. However, in vivo bone marrow transplantation experiments did not demonstrate their participation in lung vessels neomuscularization or remodeling in the CH model [27,67]. We have identified a pulmonary population of CD34^+^/CXCR4^+^/c-kit^+^ progenitor cells, which are recruited during early CH and may participate in neomuscularization [50]. Our data suggest that they are resident, as they do not express CD45. It is still difficult to define the respective role and time course of resident and circulating c-kit^+^ cells without further lineage tracing experiments. In PAH patients, increased numbers of circulating CD34^+^/CD133^+^/c-kit^+^ cells were observed in association with c-kit^+^ cells in remodeled arteries [36]. However, only rare c-kit^+^ cells co-expressed α-SMA, indicating that in patients either they are not a major contributor to SMC and myofibroblasts or that they have lost c-kit expression.

Among bone marrow-derived progenitor cells, there is a subpopulation called fibrocytes that coexpress CD34, monocyte lineage markers, such as CD45, CD14, and CD11, and fibroblast proteins collagen-I, collagen-III, and vimentin (for review on fibrocytes, see [72]). They produce extracellular matrix (ECM) components, and they can also differentiate into α-SMA^+^ myofibroblast and may participate in tissue injury during inflammation or ischemic process, aberrant healing, and angiogenesis. Fibrocytes contribute to lung fibrogenesis in several types of fibrotic diseases, including idiopathic pulmonary fibrosis, scleroderma, and pulmonary hypertension [72,73]. The mechanism underlying the recruitment of fibrocytes during vascular remodeling still needs more insight. Accumulation of stem and progenitor cells at sites of injury requires CXC chemokine receptor 4 (CXCR4), a G-protein coupled receptor for CXC chemokine ligand 12 (CXCL12) [74,75]. In PAH, Farkas et al. studied the role of CXCR4^+^ cells in the accumulation of c-kit^+^ cells in Sugen/hypoxia rats. They proposed that one of this cell’s subpopulations, expressing α-SMA, localized in and around the pulmonary lesions, could be fibrocytes [76]. Moreover, it has been shown that the number of circulating fibrocytes has significantly increased in PH individuals compared with controls [77]. Further investigations are required to explore fibrocytes as a cell population playing a key role in vascular remodeling in pulmonary hypertension.

## 4. Mechanisms for Stem/Progenitor Cell Recruitment during PH

The role of stem/progenitor cells in lung vessel homeostasis has yet to be studied. Published studies have focused on their activation, recruitment, proliferation, and differentiation in the course of PH development (Figure 2).

### 4.1. BMP

Impaired BMPR2 signaling is observed as a common feature in PAH pathogenesis [1]. This member of the TGF-β superfamily of signaling molecules represents the main susceptibility factor for PAH development [78]. Since its first discovery, numerous studies have shown the important role of the BMP pathway in controlling vascular cell function. It was first demonstrated that BMP signaling is critical to induce SMC progenitor cell differentiation during development in the gut [79] and the muscle [80]. In adult lungs, dysfunctional BMPR2 signaling is associated with β-catenin activation and results in pulmonary ABCG2^+^ mesenchymal progenitor cell expansion but without terminal differentiation as pericytes and as α-SMA^+^ SMC-like cells [81], leading to dysfunctional microvessels. Taken together, these data point to BMP as promoters of SMC differentiation that could be impaired with the deficient BMPR2 signaling that takes place in PAH. Reduced BMPR2 signaling also increases the number of circulating EPC [18] and promotes EndMT [29], the transition of PAH EC to smooth muscle-like cells [63]. Hence, although BMP signaling is essential for stem cell self-renewal and fate determination [82], studies addressing BMP regulation of pulmonary vascular progenitors are still needed.

### 4.2. TGF-β

TGF-β is a known regulator of mesenchymal stem cell proliferation and differentiation [83]. This factor plays a major role in the development of PAH pathology, and its pathway was found overactivated in vascular cells during PAH [54]. TGF-β induces CD34^+^/PDGFRα^+^ progenitor cell [50] and pericyte differentiation into SMC [54,84] through Smad2/3 phosphorylation [53]. It also induces MSC differentiation into SMC via Notch pathway activation [85]. Pericytes from iPAH patients are more responsive to TGF-β due to increased TGF-βRII receptors [53]. This may explain in part the reduced vascular remodeling after TGF-β signaling inhibition or during the expression of a dominant-negative TGF-βRII mutant [86,87]. TGF-β signaling may also probably lead to lung ABCG2^+^ mesenchymal progenitor cell differentiation into profibrotic myofibroblasts as observed in pulmonary fibrosis [84].

EPC regulation by TGF-β appears complex as on one side, it promotes EndMT in pulmonary EC clones leading to SMC-like cell production in neointima [29], and on the other side, it enhances circulating EPC angiogenic properties [88]. These various effects may depend on the balance between Smad2-dependent TGF-β and Smad1/5/9-dependent BMP signaling [29] and the inflammation [59].

### 4.3. FGF

Increased lung FGF-2 signaling has been associated with PAH. FGF-2 promotes pericyte proliferation and migration, and FGF-2 neutralizing beneficial antibody effects on PH-induced neomuscularization were, in part, due to a decreased recruitment of pericytes [54]. We have also observed that PW1^+^/CD34^+^ progenitor cells are more proliferative when treated with FGF-2 (unpublished observation). Therefore, FGF-2 induction of progenitor cell recruitment could be an important pathway for PH-associated vascular remodeling, in particular for intussusceptive angiogenesis [89].

Several studies showed that FGF10-FGFR2 signaling regulates SMC progenitor cells in the lung parenchyma during development [90,91,92]. While the expression of both proteins appears increased in iPAH [93], the role of the FGF-10 axis on PH-associated vascular remodeling and vascular progenitor cell function has not yet been evaluated.

### 4.4. Inflammatory Cytokines IL-6 and TNF-α

Inflammation is an important hallmark of PAH, and Interleukin 6 has been shown to be a major inflammatory cytokine involved in the pathology in patients and experimental models [94]. IL-6 stimulates pericyte migration but not their proliferation nor their terminal differentiation towards contractile α-SMA^+^ cells [54]. This signaling pathway contributes to induce small pulmonary vessel pericyte coverage in the mouse CH model. IL-6 could also potentiate the participation of EPC in vascular lesions as it stimulates their angiogenic potential [95].

Macrophages can induce EC differentiation of the stem/progenitor cells while simultaneously inhibiting their differentiation in SMC. TNF-α treatment of EPC increased migration and incorporation into vessel-like structures [96]. Mechanistically, both effects were mediated by macrophage-derived TNF-α via TNF-α receptor 1 and canonical nuclear factor-κB activation [97].

### 4.5. SDF-1/CXCR4/CXCR7 Pathway

As emphasized above, several stem/progenitor cell types, in particular c-kit^+^ cells, express CXCR4 while pericytes express CXCR7. SDF-1/CXCL12, a ligand for both CXCR4 and CXCR7, is a major mobilizing factor for bone marrow stem cells as EPC [98]. It is produced by pulmonary EC and macrophages, in particular, during PAH [18,99] but may also be derived from platelets [100]. Its role in mobilizing resident and circulating progenitor cells has been demonstrated in PH models [50,68,71,76,99] and patients [36], whereas CXCR7 was not involved in c-kit^+^ cell recruitment and vascular remodeling [68]. CH induces an early increase in lung SDF-1 concentration that is maintained during the course of CH models [68,76], in particular, via Hypoxia-Induced Factor (HIF)-1 and -2 induced transcriptional regulation [101,102]. SDF-1 does not induce lung resident CD34^+^/PDGFRα^+^/CXCR4^+^ progenitor cell proliferation but rather regulates their migration and/or differentiation in SMC [50] while it induces CXCR7^+^ pericyte proliferation and migration [53,99]. The role of the SDF-1 pathway on circulating EPC participation in complex vascular lesions awaits experimental evidence. Blocking SDF-1 reversed PH-associated vascular remodeling in two severe rat PH models [99], supporting an important role for this factor in regulating vascular progenitor cell recruitment and differentiation in SMC during PAH.

### 4.6. PDGF Pathway

The PDGF pathway is a central regulator of vessel structure and SMC function. Two PDGFR receptor isoforms are present on pulmonary SMC progenitor cells: the PDGFRα [50] and β [44,54], which can form homo- and heterodimers. The different PDGFs and PDGFRs have been found activated in the pulmonary vessels of iPAH patients [103], and the beneficial effect of Imatinib treatment (which inhibits PDGFR, c-kit, and c-abl) has been attributed mostly to PDGFRβ inhibition [104]. Unfortunately, although Imatinib treatment showed improvement in exercise capacity and hemodynamics in patients with advanced PAH, it also led to serious adverse effects and high treatment discontinuation and was abandoned [105]. The roles of PDGF-B and PDGFRβ were demonstrated by genetic ablation, which reduced CH-induced PH and remodeling while they were increased by PDGFRβ constitutive activation [106,107]. This pathway is a major regulator for the recruitment of pulmonary medial SMC by upregulating KLF4 transcription factor expression inducing cell dedifferentiation to allow their migration and subsequently by promoting their clonal proliferation [108]. PDGFRβ activation also led to mesenchymal Gli1^+^ progenitor cell expansion [109], which seems to be regulated by a balance between the PDGFRβ pathway and the hedgehog pathway (see below). PDGFRα activation could also be at play in regulating progenitor cells. Indeed, this receptor is also highly expressed in iPAH patient lungs [103], and its specific ligand PDGF-A regulates alveolar SMC production during lung development [110] as well as gut SMC [79]. The PDGFRα activation mediates adventitial MSC differentiation into myofibroblasts in arteriovenous fistula [111]. These results suggest the PDGFRα pathway could also be at play during PAH-associated vascular remodeling.

### 4.7. Wnt

Wnt (from Wingless in drosophila) signaling relies on canonical (via β-catenin activation) and non-canonical pathways activated by Wnt binding on Frizzled (Fzd) receptor (for review, see [112]). Wnt activation during PAH was observed as a common transcriptional signature in multiple pulmonary cell types [113], and both Wnt ligand and Wnt pathways were found upregulated in iPAH patient vessels [114,115]. During lung development, Wnt signaling activates SMC progenitor cell proliferation and is required to induce expression of PDGF receptors which are important for SMC differentiation [116]. Moreover, recent results indicated that Dickkopf WNT Signaling Pathway Inhibitor 3 (Dkk3), a Wnt modulator, can induce differentiation of aortic adventitial vascular Sca-1^+^ progenitor cells and fibroblasts into SMC via activation Wnt pathways but also of the TGF-β/ Activating Transcription Factor 6 (ATF6) pathway [117]. In addition, Dkk3 is able to signal by binding the CXCR7 receptor to stimulate these Sca-1^+^ progenitor cells’ migration [118]. Recent results on ABCG2^+^ mesenchymal cells suggest that Wnt activation prevents differentiation of these progenitor cells towards EC through β-catenin activation [119]. This is in line with other studies in which inhibition of Wnt signaling enhanced the angiogenic properties of endothelial-colony-forming cells [120] and stimulated the release of vasculogenic progenitor cells from bone marrow. On the other hand, activation of β-catenin in ES cells-derived c-kit^+^/Sca-1^+^ progenitor cells led to increased EC differentiation and decreased SMC markers [121], suggesting that Wnt signaling pathways in different progenitor cells may lead to various differentiation fates. Indeed, recent data showed that canonical β-catenin dependent Wnt signaling promotes EPC differentiation in adults [122] and during development [123]. Taken together, these results obtained suggest that circulating and pulmonary vascular progenitor cells are probably recruited by the increased Wnt signaling observed during PAH.

### 4.8. Endothelin (ET-1)

Plasma ET-1 level is increased in animal models and PAH patients in correlation with markers of severity, such as pulmonary vascular resistance [124]. Dual and selective endothelin receptor antagonists constitute a major therapeutic strategy for PAH patients. Some studies suggest that progenitor and stem cells could also be targeted by these therapeutic molecules in addition to EC and SMC. Two groups have observed that ET-1 modulates the differentiation fate of various MSC, derived from adipose tissue or bone marrow, in part through Akt activation [125,126]. In addition, ET-1 shows a strong cytoprotective effect on MSC viability [127]. ET-1 also promotes clonal expansion of ISL1^+^ vascular progenitor cells obtained from embryonic stem cells [128] and proliferation of pericytes [129]. These results suggest that ET-1 may affect lung stem and progenitor cell proliferation and differentiation during PH development. We have indeed observed that CD34^+^/PW1^+^/PDGFRα^+^ progenitor cells express ET1A receptor (unpublished observations), indicating that they could be targeted by the increased ET-1 production observed during PH.

### 4.9. Notch

Vascular progenitor cells are regulated by the Notch pathway. This signaling pathway involves direct intercellular interactions mediated by membrane-bound Notch ligands (Dll-1, Dll-4, Jag-1, Jag-2, and Dll-3) and Notch receptors (Notch 1 to 4). Following binding, Notch receptors are cleaved by A Disintegrin And Metalloproteinase domain-containing protein 10 (ADAM10) and γ-secretase releasing the notch intracellular domain (NICD) that translocates to the nucleus to act as a transcription factor (for review, see [130]). Notch stimulates the differentiation of Sca-1^+^/CD146^−^ progenitor cells into pericytes [131], of Tie1^+^ precursors in SMC [132], of PDGFRβ^+^ progenitor cells into SMC [133] and of MSC into SMC [85,134]. In addition, differentiation of circulating CD34^+^/CD31^+^ EPC depends on Notch1 [135].

The role of the Notch pathway in regulating PH-associated vascular pulmonary remodeling was demonstrated using γ-secretase inhibitors, which prevent Notch cleavage necessary for intracellular signaling [136,137,138]. However, the impact of the different Notch receptors is somewhat debated. Expressions of EC-associated Notch 1, of SMC-associated Notch3, and of plexiform lesions-associated Notch 4, are enhanced in PAH patients’ vascular lesions [136,139,140]. Notch 1 and 3 were also found enhanced in PAH experimental models, CH mice [136] and MCT rats [137], Sugen/hypoxia [140,141]. The Notch pathway is activated within weeks 1-2 of CH, leading to pulmonary vascular remodeling [142] and increased *Hes/Hey* family target genes expression [137]. Interestingly, in the study of Steffes et al. [138], Notch 3 expression was restricted to some specific SMC in pulmonary arterioles which participate in neointima formation. This is reminiscent of the scarce activated SMC observed by Sheikh et al. in the pulmonary arteriolar wall during CH-induced remodeling that participates in neomuscularization [44]. It will be very informative to investigate whether these observations indeed involve the same type of primed SMC.

### 4.10. Metabolism

In adults, several quiescent stem cells (hematopoietic, neural, muscle, hair follicle, bone marrow MSC) exhibit a metabolism based on a high glycolytic activity to provide energy. Differentiating recruited stem cells appear to shift to oxidative phosphorylation with increased mitochondrial density [143]. However, other stem cells that are permanently activated, such as intestinal stem cells or spermatogonia, instead seem to rely on mitochondrial metabolism. Lung stem/progenitor cells metabolic behavior during PH is still to be determined, but some lines of evidence suggest that it is probably altered as shown for other cells. Thus, increases in aerobic glycolysis and upregulation of the pentose phosphate pathway were observed in *BMPR2* mutant human pulmonary microvascular endothelial cells [144]. A Warburg-like glycolytic reprogramming has been observed in pulmonary mesenchymal cells during PH, with increased glycolysis [145], activation of the pentose phosphate pathway (for review, see [146]), and is associated with alterations in proliferation, phenotypic changes, apoptosis. This could be linked to HIF-1α activation, an important player in PH development [147,148] and a major inducer of glycolysis vs. oxidative phosphorylation.

Glucose-6-phosphate dehydrogenase (G6PD), the first enzyme of the pentose phosphate pathway, may act as a link between reprogrammed metabolism and aberrant gene regulation, but its role during PH is controversial. G6PD expression appears downregulated in iPAH patients’ lungs, and G6PD deficiency leads to PH development in mouse [149]. On the contrary, other studies showed that G6PD expression is increased in iPAH patients and PH models [150,151] and that G6PD deficiency prevents PH development [152] through epigenetic control of vascular cells. Variations in resident stem/progenitor cell metabolic status during PH is still unknown, but Chettimada et al. demonstrated that circulating hypoxic CD133^+^ progenitor cell proliferation is dependent on an increased G6PD activity which promotes their dedifferentiation [151]. Indeed, G6PD controls the chromatin modifications by regulating histone deacetylase activity, and its inhibition enhances the expression of SMC-restricted genes while decreasing the expression of the stem/progenitor marker Sca-1 in cultured SMCs [153].

Another important pathway linked to glucose metabolism is the O-GlcNAcylation, i.e., the glycosylation of intracellular proteins with N-acetyl-d-glucosamine, which was found to be increased both in iPAH patients and in PH models [154]. This dynamic post-translational modification leads to changes in protein function, trafficking, and localization and regulates stem cell self-renewal, pluripotency, and differentiation, in part, via epigenetic regulations (for review, see [46]). In particular, O-GlcNAcylation stabilizes HIF-1α [155], SMAD4, a major regulator of TGF-β and BMP signaling pathways [156], or the EndMT regulator SNAIL1 [157]. Therefore, modifications in O-GlcNAc transferase activity and substrate level (glucose) may alter stem/progenitor cell function and participation in vascular remodeling. Hence, TGF-β induction of aortic c-kit^+^ cell differentiation into SMC was dependent on increased O-GlcNAcylation, with SRF and myocardin, the key transcription factors for SMC differentiation, being two of the modified proteins [70].

Other important metabolism-linked proteins could also participate in progenitor/stem cell regulation, such as Adenosine Monophosphate-Activated Protein Kinase (AMPK). AMPK is known as a sensor of metabolic stress, which detects energetic disequilibrium, and it has been shown to regulate EPC differentiation into EC [158]. In addition, AMPKα2 deficiency leads to increased PH severity in a mouse CH experimental model [159].

## 5. Therapeutic Use of Stem/Progenitor Cells

### 5.1. Endothelial Progenitor Cells

EPC could be helpful diagnostically as circulating biomarkers for predicting risks, may provide a source of vascular progenitor cells to facilitate neovascularization, and could be manipulated in vitro to enhance their ability for vascular repair (Figure 3).

As mentioned previously, numerous studies have established differences in the number and the phenotype of EPC between PAH and control patients; levels of circulating EC and EPC seem to be elevated in PAH patients, and the number of these cells could be positively correlated with pulmonary artery pressure [18,20,160,161], but other works reported opposite observations [11,12,13,14,15]. Therefore, the relevance of EPC continues to be discussed as a potential biomarker for the PAH, and this is closely related to the choice of markers used to discriminate these endothelial cell populations [162,163] and also to the clinical PH case studied [164,165]. Moreover, circulating microparticles released by vascular EC are increased in PAH patients, and their levels seem to be correlated with pulmonary vascular resistance and predict a poor outcome [166,167,168]. These microparticles are membrane-shed submicron vesicles released by altered EC, bearing endothelial surface markers and which can modulate cellular function. In addition, PH can be induced in healthy mice by injecting PH mouse-derived-extravesicles, raising the question that they could mediate the disease [169]. Thus, EC and EPC microparticles may be considered as biological markers and have importance in the pathogenesis of PAH (for review, see [170]).

Alongside that, several investigations have shown that intravenous EPC transplantation had beneficial effects in monocrotaline-induced PH models [171,172,173,174]. In mouse, BM-derived cells limit pulmonary vascular remodeling induced by vascular injury but not by hypoxia [175]. In addition, other studies have shown that transfer of VEGFR-2^+^/Sca-1^+^/CXCR4^+^ cultured early-outgrowth EPCs failed to reverse CH-induced PH [27] and that BM-derived late-outgrowth endothelial cell can contribute to vascular repair of an injured systemic artery, but they cannot rescue injured pulmonary vasculature under MCT-induced PH [176]. Likewise, BM-derived cell injection had no effects on pulmonary hypertension in the pneumonectomized rats [66]. These publications suggest that the sources of the transplanted cells and the PH study models (hypoxia versus monocrotaline) may account for their differences in incorporation rate into the pulmonary endothelium and their beneficial effects.

Thus, EPC has proven to be an endless reservoir of manipulation in order to take full advantage of their regenerative capacity, sometimes by combining them with other treatments, and to prevent their early death post-transplantation [177,178,179]. In rats, preconditioning of EPC with pinocembrin, a flavonoid naturally abundant in propolis, leads to an increase in nitric oxidase and VEGF production by EPC, and their transplantation reduces monocrotaline-induced inflammation, the medial wall thickness, and ameliorates the endothelial function [180]. EPC combined with sildenafil showed better results than EPC treatment alone [181]. Nagaya et al. reported that adrenomedullin DNA-transduced EPC improves pulmonary hemodynamics and increases survival rate compared with EPC transplantation alone [172]. Interruption of the CD40 pathway in EPC increases their incorporation in the intima and might reduce their secretion of inflammatory cytokines [182]. In a hyperkinetic PH rabbit model, EPC transfected with human HIF-1α are more efficient at reversing the hypertrophy of the right ventricle, pulmonary remodeling, and increasing the number of small pulmonary arteries than EPC transplantation alone [183]. In human, in the Pulmonary Hypertension and Angiogenic Cell Therapy (PHACeT) trial, patients with stable severe PAH received EPC overexpressing eNOS in order to restore the vascular homeostasis and endothelial function. They showed a good tolerance and short-term hemodynamic improvements [184].

Moreover, despite short retention time in lungs and poor efficacy for long-term in the pulmonary vasculature, BMPR2-augmented EPC trigger upregulation of BMPR2 in the pulmonary vessels and ameliorate monocrotaline-induced PH in rats because they might release exosomes expressing VEGFR-2, CD54, and CD105, suggesting that future investigations using purified exosomes alone may be relevant to understand the mechanism [185] and could represent advantages over EPC/EC-based therapy [186,187] (for review on exosomes, see [188]). This supports a beneficial paracrine activity of EPC in addition to their incorporation into pulmonary vessels as observed in MCT-treated rats [173].

### 5.2. MSC Therapy

MSC have aroused great interest as therapy for the past 20 years. These non-hematopoietic multipotent cells can be easily isolated from bone marrow, adipose tissue, peripheral blood, or umbilical cord blood. They display large differentiation capacities (at least in vitro), immunomodulatory properties, important paracrine activity, and are weakly immunogens due to lack of class II histocompatibility complex antigens [48]. Rodent and human MSC administration in rodent PH models have proven the interesting therapeutic properties of these cells in reversing vascular remodeling, mPAP increase, pulmonary inflammation, and right ventricular hypertrophy (for review, see [189]). Careful examination of recipient lungs revealed that very few MSC engrafted, suggesting that the beneficial effects could be mainly paracrine [190]. Indeed, several authors have recently shown that MSC-derived exosomes could attenuate PAH development in pre-clinical models [191,192]. Various studies also showed that modifying MSC could potentiate their effect: e.g., by stimulating them with sphingosine 1 phosphate or transducing them with eNOS, prostacyclin synthase, or heme oxygenase-1 expression vector [193,194,195]. As determined by searching the clinicaltrials.gov (accessed on 26 April 2021) database, these encouraging results led to only a few clinical trials evaluating the beneficial effect of MSC treatment in PH patients.

## 6. Conclusions and Perspectives

Growing evidence stated here and by others [38] establishes circulating and resident progenitor and stem cells as important contributors in the vascular remodeling accompanying PH. The lack of specific markers led many groups to propose their own set of markers, and we thus need unifying studies to allow comparisons of the various cells involved. Another major drawback of progenitor/stem cell studies is that the link between the different cell populations identified has not yet been analyzed. So far, it is not known whether they are hierarchically connected or whether they belong to unrelated cell populations. For example, transcriptomic studies in the heart suggested that c-kit^+^ cells were undifferentiated, whereas Sca-1^+^ cells were closer to cardiomyocytes [196]. However, resident progenitor cell fate is closely linked to the tissue in which they live. Hence, their transitional state and markers and terminal differentiation may be different in the lung. Indeed, lineage tracing approaches in the context of PH are lacking to prove the involvement of the different stem and progenitor cells. In particular, lineage tracing investigations in mouse PH models with more severe vascular remodeling, such as House Dust Mite mouse model [138], left pneumonectomy/monocrotaline pyrrole mouse model [58], or mice with a EC specific-loss of prolyl-4 hydroxylase 2 (PHD2) [197], could help us identify the fate of progenitor cells and dissect the hierarchy between the different cell populations while getting closer to human pathology pattern. In addition, genome editing advances using CRISPR/Cas9 promote new opportunities to generate Cre reporter rats for lineage tracing studies in MCT and Sugen/hypoxia rat models.

Pathways regulating the recruitment of vascular progenitor/stem cells associated with PH development have also been deciphered. This field may unveil new important pathways that could be targeted in patients (such as humoral factors, extracellular matrix composition, or mechanical stresses). Interestingly, the role of the Sonic hedgehog (SHH) signaling pathway in pulmonary hypertension development has not been investigated yet, but several results point to a role in regulating vascular progenitor cell function that could be involved in vascular remodeling. First, SHH directly promotes circulating EPC proliferation, migration, adhesion, and tube formation [198,199]. Second, Gli1, considered by several authors as a marker for SMC or myofibroblast progenitor cells [200], is a direct effector and target of SHH signaling pathway, and its expression coincides with SHH activated regions and regulates vascular progenitor cell fate and recruitment [109,134]. Third, Sca-1^+^ vascular progenitor cells are clustered in a zone SHH signaling in the adventitial layer of systemic arteries [201], where SHH may regulate their proliferation, self-renewal, and survival [134,201]. Conversely, the recent discoveries of new genes with deleterious variants leading to PAH susceptibility, such as T-Box Transcription Factor 4 (*TBX4*) and SRY-Box Transcription Factor 17 (*SOX17*) [78], or to PVOD, such as General Control Nonderepressible 2 (*GCN2*) [202], are offering new leads to understand the regulation of progenitor/stem cells during the course of the pathology.

## Figures and Tables

**Figure 1 cells-10-01338-f001:**
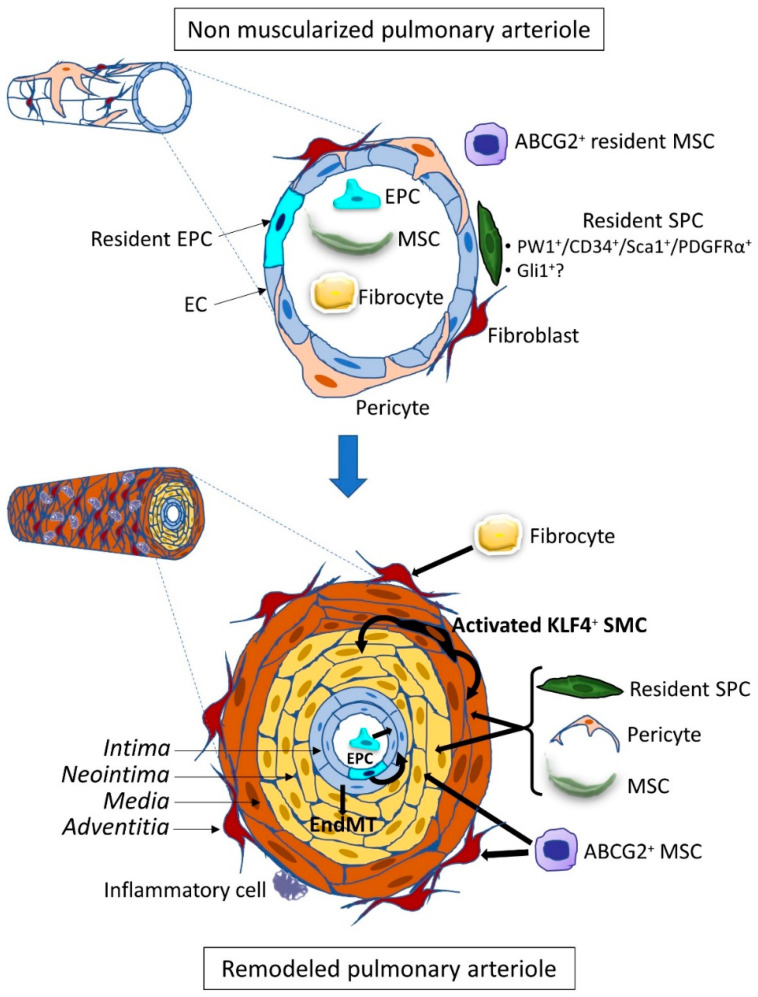
Role of Stem/progenitor cells in pulmonary hypertension-associated vascular remodeling. Pulmonary hypertension is characterized by excessive pulmonary vascular occlusive remodeling with an increased medial thickness of normally muscularized arterioles and muscularization of previously non-muscularized arterioles. This remodeling involves the production of new endothelial cells, myofibroblasts (MF), vascular smooth muscle cells, fibroblasts, and extracellular matrix changes leading to the formation of a neointima between the endothelium and the internal elastic lamina, to medial hypertrophy and vascular/perivascular fibrosis and inflammation. Circulating (EPC and MSC) and resident vascular stem/progenitor cells (EPC, activated SMC, SPC, pericytes, MSC) were identified. These cells can be either mobilized from cellular niches within or close to the vessel wall or in the lung interstitium, or from distant tissues, mainly the bone marrow, through the circulation. In advanced lesions, EC are able to undergo an endothelial–mesenchymal transition (EndMT) and to transdifferentiate into SMC-like cells. Thick black arrows represent differentiation fates of progenitor/stem cells.

**Figure 2 cells-10-01338-f002:**
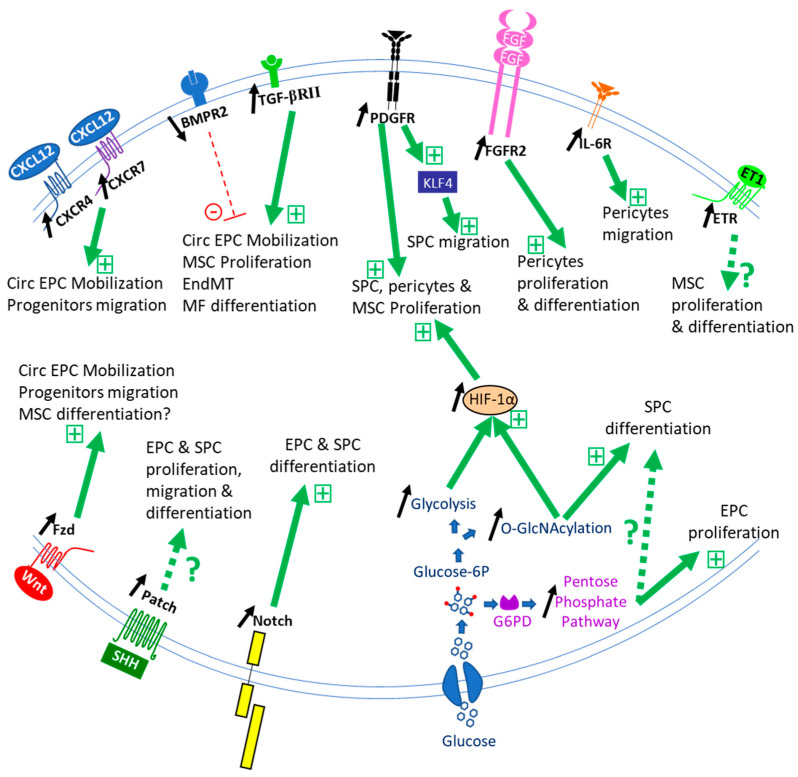
Schematic representation of the main known signaling mechanisms for stem/progenitor cell recruitment during PH. Receptors (CXCR4, CXCR7, BMPR2, TGF-βR, PDGFR, FGFR2, IL-6R, ETR, Fzd, Patch, Notch) and ligands (CXCL12/SDF-1, BMPs, TGF-β, PDGFs, FGF-2, ET-1, Wnt, SHH, Notch ligands, respectively) are represented at the membrane of progenitor/stem cells regulating their mobilization, proliferation, migration, and/or differentiation during PH development. The balance between BMP and TGF-β signaling is disrupted, leading to progenitor/stem cell recruitment and participation in remodeling. Aberrant HIF-1α activation enhances the proliferative responses and differentiation of SPC, SMC, and pericytes to mitogens, such as PDGF and FGF-2 via PDGFR and FGFR2 receptors. Disrupted glucose metabolism with an increase in the O-GlcNAcylation, the glycolysis, and the pentose phosphate pathways could induce progenitor/stem cell recruitment and also participate in the stabilization of HIF-1α. BMP and TGF-β receptors are multimodal receptors consisting of various combinations of type I and type II. BMPR2 and TGF-βRII are represented here. ETR = ETA or ETB receptors. Black arrows represent the increased or decreased activity of pathways observed during PH development. Thick green arrows represent the positive effect of pathways on progenitor/stem cell recruitment or function. Dotted green arrows represent potential regulatory pathways derived from studies in other systems. The red dotted line represents a reduced inhibitory effect of the BMPR2 pathway on progenitor/stem cell recruitment or function.

**Figure 3 cells-10-01338-f003:**
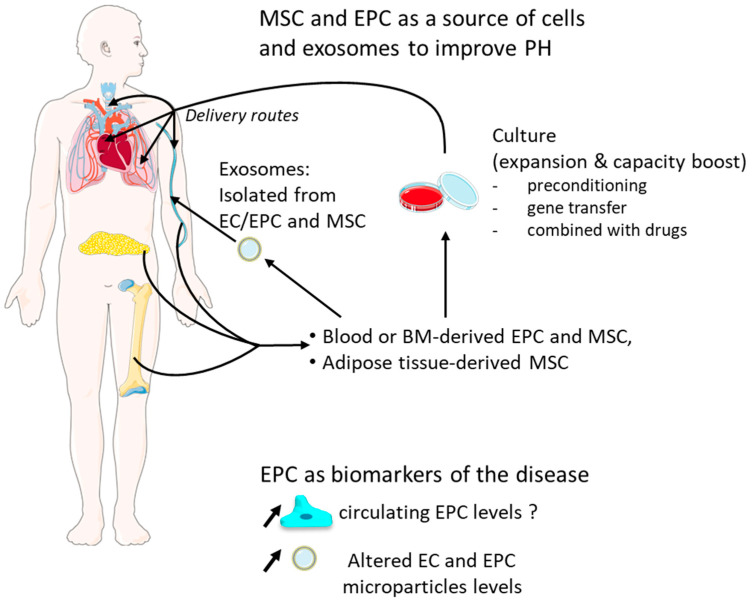
Schematic illustration of stem/progenitor cells potential use for PH therapy. EPC could be helpful diagnostically as circulating biomarkers for predicting risks, but their relevance continues to be discussed. Circulating microparticles released by altered EC are increased in PAH patients, and it seems to be correlated with pulmonary vascular resistance and predict a poor outcome. MSC can be easily isolated from bone marrow, adipose tissue, peripheral blood, or umbilical cord blood as well as EPC from bone marrow and peripheral blood. EPC, MSC, and derived exosomes can be manipulated in order to boost their regeneration capacity and then injected via different delivery routes to improve PH.

## Data Availability

No new data were created or analyzed in this study. Data sharing is not applicable to this article.

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
