# Peer review of "Progenitor/Stem Cells in Vascular Remodeling during Pulmonary Arterial Hypertension"

_cells, 2021, doi:10.3390/cells10061338_

Round 1
Reviewer 1 Report
This is a well-written review that will increase our current understanding of progenitor cells contribution to the pulmonary vascular remodeling and pathogenesis of PAH.
1. The authors should raise these kinds of discussions in the way we study progenitor cells in vascular remodeling in PAH. There are limitations in most current published studies in progenitor cell studies in PAH. Firstly, most of the studies did not use a lineage tracing approach to identify the cell fate of progenitor cells. Secondly, most of the animal models (hypoxia or hypoxia/SuHx mice) the PAH folks used did not develop severe vascular remodeling. There were severe mouse models with severe vascular remodeling including Egln1Tie2Cre mice (PMID: 27143681) , House Dust Mite (HDM) Mouse Model (Ref 137), left pneumonectomy plus monocrotaline pyrrole mouse model (PMID: 24201301), which can be used for lineage tracing. Alternatively, as CRISPR/Cas9 technology develops, it is easier to generate rats for lineage tracing in monocrotaline and SuHx/hypoxia-induced PH models.
2. It is highly recommended to add a Perspective section about the further direction of progenitor cell study in PAH. As Hedgehog signaling has not been explored in PAH, it is considered to move to this section.
3. Line 194-195, in addition to the factors listed, a recent study also showed that many other EC-derived factors including CXCL12, ET-1, and MIF induce SMC proliferation. These factors should be included. (PMID: 29664678 )
4.KLF4 should be included in figure 2.
Author Response
We thank the reviewer for these comments that led to improve the manuscript.
- Comment 1. We have changed the conclusion part to Conclusion and perspectives in which we have highlighted the need for lineage tracing models that can be used with severe model of PH as suggested by reviewer 1. The new sentences are written in blue. We added p20:
Indeed, lineage tracing approaches in the context of PH are lacking to prove the involvement of the different stem and progenitor cells. In particular, lineage tracing investigations in mouse PH models with more severe vascular remodeling, such as House Dust Mite mouse model [138], left pneumonectomy/monocrotaline pyrrole mouse model [58] or mice with a EC specific-loss of PHD2 (prolyl-4 hydroxylase 2) [197], could help us identify the fate of progenitor cells and dissect the hierarchy between the different cell populations, while getting closer to human pathology pattern. In addition, genome editing advances using CRISPR/Cas9 promote new opportunities to generate Cre reporter rats for lineage tracing studies in MCT and Sugen/hypoxia rat models.
- Comment 2. We moved the paragraph on the SHH pathway (which was shortened) to the conclusion and perspectives section p20
- Comment 3. We added a sentence and a reference on SMC regulation by EC-produced SDF-1 and MIF p6:
The increased EC expression of SDF-1 (Stromal cell-Derived Factor 1) and MIF (Macrophage migration Inhibitory Factor) can induce marked FoxM1 expression in SMC inducing their proliferation as well [1,42]
- Comment 4. Figure 2 was modified to add KLF4
Reviewer 2 Report
This review article is interesting and well written.
There are several minor comments.
The abbreviations should be fully spelled at the first appearance , such as VEGHR FGF and so on.
Page 7, iPAH and hPAH → What are i and h?
Page 8, What are GFP+ cells?
Page 12 in PDGF paragraph, different actors → different factors?
Page 16 in the middle, in cultured cultures SMC→ in cultured SMCs
Page 16, near the bottom, EPC differentiation in EC → EPC differentiation into EC (?)
Page 18, Line 2, microparticles → What are microparticles?
Page 19, line 7 from the bottom, However, resident progenitor cells are tissue-specific and this may not be applied in the lung. → Why this may not be applied in the lung?
Author Response
We thank the reviewer for pointing at us unclear points of the manuscript
- Abbreviations have been fully spelled on their first appearance.
- We have removed the terms GFP+, factors, cultures which were not necessary and unclear and we corrected EPC differentiation into EC p16.
- We have explained the term microparticles p18 as :
These microparticles are membrane-shedded submicron vesicles released by altered EC, bearing endothelial surface markers and which can modulate cellular function.
- We have modified the sentence p19 to be more explicit:
However resident progenitor cells fate is closely linked to the tissue in which they live, hence their transitional state and markers and terminal differentiation may be different in the lung.
Reviewer 3 Report
General comments:
The authors summarize the current knowledge on the different progenitor and stem cells that participate in pulmonary vascular lesions, on the pathways regulating their recruitment during pulmonary arterial hypertension, and the therapeutic potential of circulating endothelial progenitor cells and mesenchymal stem cells in pulmonary arterial hypertension.
Minor concerns:
- Abbreviations, pages 58 and 60: Please add the full name of BMP and BMPR2 and check all the abbreviations throughout the manuscript.
- 200: Please add “45, 518-521. doi: 10.1038/ng.2581”.
Author Response
We thank the reviewer for these necessary corrections.
- Abbreviations have been fully spelled on their first appearance.
- We have corrected ref 200 (now 202) and added DOI to all references where they were missing.